# Biallelic loss-of-function in *NRAP* is a cause of recessive dilated cardiomyopathy

**Juha W. Koskenvuo**[1]*, **Inka Saarinen**[1], **Saija Ahonen**[1], **Johanna Tommiska**[1],
**Sini Weckström**[2], **Eija H. Seppälä**[1], **Sari Tuupanen**[1], **Tiia Kangas-Kontio**[1],
**Jennifer Schleit**[1], **Krista Heliö**[1], **Julie Hathaway**[3], **Anders Gummesson**[4], **Pia Dahlberg**[5],
**Tiina H. Ojala**[6], **Ville Vepsäläinen**[7], **Ville Kytölä**[1], **Mikko Muona**[1], **Johanna Sistonen**[1],
**Pertteli Salmenperä**[1], **Massimiliano Gentile**[1], **Jussi Paananen**[1], **Samuel Myllykangas**[1],
**Tero-Pekka Alastalo**[3], **Tiina Heliö**[2]

1 Blueprint Genetics, a Quest Diagnostics Company, Espoo, Finland, 2 Heart and Lung Center,
Helsinki University Hospital, University of Helsinki, Helsinki, Finland, 3 Blueprint Genetics Inc, a Quest
Diagnostics Company, Seattle, Washington, United States of America, 4 Department of Clinical Genetics
and Genomics, Sahlgrenska University Hospital, Gothenburg, Sweden, 5 Department of Cardiology,
Sahlgrenska University Hospital, Gothenburg, Sweden, 6 Department of Pediatric Cardiology, Helsinki
University Hospital and University of Helsinki, Helsinki, Finland, 7 Heart Center, Kuopio University Hospital,
Kuopio, Finland

* juha.koskenvuo@blueprintgenetics.com

pone.0245681

University Wexner Medical Center, UNITED
STATES

**Data Availability Statement:** There are both ethical
and legal restrictions on sharing the full data set.
Data is owned by Blueprint Genetics, which has no

## Abstract

### Background

Familial dilated cardiomyopathy (DCM) is typically a monogenic disorder with dominant
inheritance. Although over 40 genes have been linked to DCM, more than half of the patients
undergoing comprehensive genetic testing are left without molecular diagnosis. Recently,
biallelic protein-truncating variants (PTVs) in the nebulin-related anchoring protein gene
(*NRAP*) were identified in a few patients with sporadic DCM.

### Methods and results

We determined the frequency of rare *NRAP* variants in a cohort of DCM patients and control
patients to further evaluate role of this gene in cardiomyopathies. A retrospective analysis of
our internal variant database consisting of 31,639 individuals who underwent genetic testing
(either panel or direct exome sequencing) was performed. The DCM group included 577
patients with either a confirmed or suspected DCM diagnosis. A control cohort of 31,062
individuals, including 25,912 individuals with non-cardiac (control group) and 5,150 with
non-DCM cardiac indications (Non-DCM cardiac group). Biallelic (n = 6) or two (n = 5)
*NRAP* variants (two PTVs or PTV+missense) were identified in 11 unrelated probands with
DCM (1.9%) but none of the controls. None of the 11 probands had an alternative molecular
diagnosis. Family member testing supports co-segregation. Biallelic or potentially biallelic
*NRAP* variants were enriched in DCM vs. controls (OR 1052, p<0.0001). Based on the fre-
quency of *NRAP* PTVs in the gnomAD reference population, and predicting full penetrance,
biallelic *NRAP* variants could explain 0.25%-2.46% of all DCM cases.

consent to share the data so that individuals can be recognized. WES data is each individual's genetic fingerprint and provide information allowing to distinguish each person in the world by any other individual, thus the data can't be legally shared as full. Also the ethical permit acquired for this study does not allow evaluation of non-cardiac genes from any of the individuals, thus the WES data can't be shared. However, all relevant data necessary to replicate the study's results are within the paper and its Supporting Information files.

**Funding:** This work was supported by grants from the Finnish Foundation for Cardiovascular Research (TH), Aarne Koskelo Foundation (TH), Special Governmental Subsidy (EVO) grants (TH). The funders had no role in study design, data collection and analysis, decision to publish, or preparation of the manuscript.

**Competing interests:** Drs. Koskenvuo, Saarinen, Ahonen, Tommiska, Seppälä, Tuupanen, Kangas-Kontio, Schleit, Hathaway, Kytölä, Muona, Sistonen, Salmenperä, Gentile, Paananen, Myllykangas, Alastalo are full-time employees of Blueprint Genetics, a Quest Diagnostics Company, which offers genetic diagnostics for cardiomyopathies. The funder provided support in the form of salaries for authors [JWK, IS, SA, JT, ST, TKK, JS, JH, VK, MM, JS, PS, MG, JP, SM, TPA], but did not have any additional role in the study design, data collection and analysis, decision to publish, or preparation of the manuscript. The specific roles of these authors are articulated in the 'author contributions' section. All other authors have reported that they have no relationships relevant to the contents of this paper to disclose.

## Conclusion

Loss-of-function in *NRAP* is a cause for autosomal recessive dilated cardiomyopathy, supporting its inclusion in comprehensive genetic testing.

## Introduction

Dilated cardiomyopathy (DCM) is characterized by left ventricular enlargement and systolic dysfunction in the absence of other etiological causes [1]. It is typically an adult-onset disease but disease onset may take place as early as in infancy. Genetic DCM has incomplete, age-dependent penetrance and presentation may vary even within the same family ranging from asymptomatic to end-stage heart failure and sudden cardiac death (SCD). The prevalence of DCM in the general population is estimated int the range of 1:500 to 1:3,000 [2–4].

Familial DCM is typically considered to be a monogenic disorder following most commonly an autosomal dominant pattern of inheritance [1,2,5]. However, X-linked, recessive and mitochondrial inheritance patterns have been observed [6]. As much as 30–50% of DCM is thought to be genetic or familial [6,7]. Over 40 genes encoding proteins of cytoskeleton, sarcomere, nuclear envelope, ion channels, and intercellular junction such as *TTN*, *LMNA*, *MYH7*, *FLNC*, *DSP*, *TNNT2*, *RBM20*, *DES*, *TPM1* and *DMD* contribute to the monogenic forms of DCM [7–11].

Recently, biallelic protein-truncating variants (PTVs) in the nebulin-related anchoring protein gene (*NRAP*) have been identified in a few patients with severe sporadic DCM [12–15], and have been proposed to cause low-penetrant recessive DCM (Table 1). However, two healthy individuals (age 33 and 35) in these families had the same homozygous PTV, which was considered to partially question the variants' pathogenicity. *NRAP* is not yet officially a morbid OMIM gene and has not yet been curated by ClinGen (NIH) or Genomics England PanelApp [16]. Thus, it is absent from most commercially available gene panels at the moment.

Since both enrichment and co-segregation of *NRAP* variants in DCM are unknown, our aims were to 1) evaluate whether patients who underwent genetic testing due to DCM have a higher frequency of *NRAP* variants compared to controls, 2) to study co-segregation of the *NRAP* variants, and 3) to define genotype-to-phenotype associations in *NRAP*-associated cardiomyopathy.

## Materials and methods

### Patients

The cohort represents 31,639 consecutive patients referred to genetic testing relying either on whole exome sequencing platform (WES; n = 24,630) or 4,600 gene high-quality next generation sequencing assay (HQSA; n = 7009) after January 2017. The inclusion criteria for DCM group (see later) was referral to genetic testing due to diagnosis or clinical suspicion of DCM.

This registry study complies with the Declaration of Helsinki. Patients who consented for Blueprint Genetics to contact them in relation to future research findings after initial testing, were contacted through their referring healthcare professional when possibly diagnostic biallelic variants in *NRAP* gene were found in sequence data. Patients living in the Helsinki University Hospital (HUS) region in Southern Finland were recruited to the Inherited Cardiomyopathies Study or KidCMP Study, and segregation studies were carried out when possible.

**Table 1. Previously reported patients with biallelic truncations in *NRAP*.**

| | Chromosomal position | Transcript; exon | Variant | GnomAD allele frequency | Pheno-type | Age at onset (years) gender | LVEDD (mm) | EF (%) | Other clinical features | References |
|---|---|---|---|---|---|---|---|---|---|---|
| 1 | 10:115355414 | NM_198060.3 Exon 38/42 | c.4504 C>T, p. (Arg1502*)[#] | 92/282804 | DCM | 26 M | 71 | 15 | Biventricular heart failure after prolonged viral-like illness, ventricular tachycardia, CK 68 U/l (normal: 39–308), NT-proBNP 5154 pg/ml, and TnT 15.04 ng/ml. 36- year old brother[#] is healthy. | Truszkowska 2017 (12) |
| 2 | 10:115413838–115413845 | NM_001261463 Exon 5/42 | c.400_407del,p. (Cys134Serfs*12)[#] | 0/251404 | DCM | NA | NA | NA | NA | Monies 2017 (13) |
| 3 | 10:115400070 | NM_001261463 Exon 14/42 | c.1344T>A, p. (Tyr448)*[#] | 34/263614 | DCM | 3.5 F | >3SD | 15 | DCM after mild respiratory viral infection, suspected myocarditis, LVEDV 245ml/m$^2$. Died before planned Htx. In autopsy no signs of myocarditis. | Vasilescu 2018 (14) |
| 4 | 10:115413838–115413845 | NM_001261463 Exon 5/42 | c.400_407del,p. (Cys134Serfs*12)[#] | 0/251404 | DCM | 1 F | | 15 | Cardiac arrest. CK 163 U/l (normal 26–168), Brother died at 17 months for cardiomyopathy, genotype unknown. Father[#] is healthy | Ahmed 2019 (15) |

Abbreviations: DCM, dilated cardiomyopathy; F, female; GnomAD; Allele frequency in gnomAD, Htx, heart transplantation; LVEDD; Left ventricular end-diastolic diameter; M, male; NA, not available; Pheno, phenotype

[#], homozygous. No homozygotes for these variants are present in the gnomAD reference population cohort.

Participants of the Inherited Cardiomyopathies study gave written informed consent, and the study was approved by the Ethical Review Committee of The Department of Medicine, University of Helsinki (Dnro 307/13/03/01/2011, HUS/3225/2018, TMK11§274,16.12.2015). This study has permission from Statistics Finland and Ministry of Social Affairs and Health to obtain clinical data from deceased patients for research purposes.

## Sequencing

Sample preparation including DNA isolation, fragmentation, library preparation techniques, bioinformatics, and quality control were similar for both WES and HQSA.

When required, the total genomic DNA was extracted from the biological sample using bead-based method. DNA quality and quantity were assessed using electrophoretic methods. After assessment of DNA quality, qualified genomic DNA sample was randomly fragmented using non-contact, isothermal sonochemistry processing. Sequencing library was prepared by ligating sequencing adapters to both ends of DNA fragments. Sequencing libraries were size-selected with bead-based method to ensure optimal template size and amplified by polymerase chain reaction. Regions of interest (exons and intronic targets) were targeted using hybridization-based target capture method. The quality of the completed sequencing library was controlled by ensuring the correct template size and quantity and to eliminate the presence of leftover primers and adapter-adapter dimers. Ready sequencing libraries that passed the quality control were sequenced using the Illumina's sequencing-by-synthesis method using paired-end sequencing (150 by 150 bases). Primary data analysis converting images into base calls and associated quality scores was carried out by the sequencing instrument using Illumina's proprietary software, generating CBCL files as the final output.

Base called raw sequencing data was transformed into FASTQ format using Illumina's software (bcl2fastq). Sequence reads of each sample were mapped to the human reference genome (GRCh37/hg19). Burrows-Wheeler Aligner (BWA-MEM) software was used for read alignment. Duplicate read marking, local realignment around indels, base quality score recalibration and variant calling were performed using GATK algorithms (Sentieon) for nuclear DNA. Variant data was annotated using a collection of tools (VcfAnno and VEP) with a variety of public and private variant databases including but not limited to gnomAD, ClinVar and HGMD. The median sequencing depth and coverage across the target regions for the tested sample were calculated based on MQ0 aligned reads. The sequencing run included in-process reference sample(s) for quality control, which passed our thresholds for sensitivity and specificity. The patient's sample was subjected to thorough quality control measures including assessments for contamination and sample mix-up.

Analysis in this study was limited to single-nucleotide variants, and small insertions/deletions and their combinations (INDELs) up to 220 bps within protein coding exons and exon-intron boundaries (± 20 bps). Copy number variations were excluded from the analysis. Performance metrics were as follows: WES: Median coverage 174x, >20x depth at target region 99.4%, >20x depth at *NRAP* gene 100%, sensitivity for SNVs 99.65%, indels 1–50 bps 99.1%, and specificity >99.9% and HQSA: median coverage 143x, >20x depth at target region 99.86%, >20x depth at *NRAP* gene 100%, sensitivity for SNVs 99.89%, indels 1–50 bps 99.2% and specificity >99.9%). Both assays have been validated in a CAP and ISO accredited laboratory (Blueprint Genetics, Finland).

***NRAP* variants.** Since our aim to evaluate the role of potentially disease causing *NRAP* variants, the analysis was limited only to the variants with the highest potential to cause disease, specifically PTVs (nonsense, frameshift, canonical splice site, start lost) and missense variants as most of the synonymous and intronic variants are less likely to be disease causing. In addition, variants were included into further analysis only if no homozygous carriers were present in the Genome Aggregation Database control cohort (gnomAD) [17] and missense variants with 100 or less heterozygous individuals in gnomAD. Frequency of such high-quality variants were compared between patients with clinical or suspected dilated cardiomyopathy (DCM group), other cardiology indication (Non-DCM cardiac group consisting patients tested due diagnosis or suspicion inherited aortopathy, channelopathy or cardiomyopathy other than DCM) or any other clinical indication for panel or exome testing (Control group).

## Statistics

Comparisons between groups were performed with either Fisher's exact or Chi-Square test for categorical variables and unpaired T-test for normally distributed continuous variables. Odds ratios (ORs) for DCM and non-DCM cardiac group *vs*. control group were calculated, and 95% confidence intervals (CIs) were determined using the conditional maximum likelihood/Fishers' method. Normally distributed parameters are presented as mean ± standard deviation.

## Results

### Whole exome sequencing (WES) data set and *NRAP* variants

All variant calls from the *NRAP* gene were queried from the internal variant database in 31,639 individuals who underwent genetic testing using NGS-panels or direct WES approach. Of these patients, 577 were tested due to DCM or suspected DCM (DCM group), 5,150 due to suspicion of other monogenic cardiac disease (Non-DCM cardiac group) and 25,912 served as controls (control group).

**Table 2. *NRAP* variants observed in the patients with dilated cardiomyopathy.**

| Patient | Variants | HGVS nomenclature | Variant type | Exon | gnomAD AC | SIFT | Cons. | ACMG Class |
|---------|----------|-------------------|--------------|------|-----------|------|-------|------------|
| **1** | **10:115374685** | **c.3099G>A, p.(Trp1033*)** | **Nonsense** | **28/42** | **18** | | | **LP** |
| **2** | **10:115356904** | **c.4371del, p.(Thr1458Glnfs*36)** | **Frameshift** | **37/42** | **100** | | | **P** |
| | **10:115423570** | **c.72G>C, p.(Gln24His)** | **Missense** | **1/42** | **34** | **Delet. (0.01)** | **Full** | **P** |
| 3 | 10:115356904 | c.4371del, p.(Thr1458Glnfs*36) | Frameshift | 37/42 | 100 | | | P |
| | 10:115423570 | c.72G>C, p.(Gln24His) | Missense | 1/42 | 34 | Delet. (0.01) | Full | P |
| 4 | 10:115400070 | c.1344T>A, p.(Tyr448*) | Nonsense | 14/42 | 35 | Delet. (0.01) | Full | P |
| | 10:115423593 | c.49G>A, p.(Glu17Lys) | Missense | 1/42 | 5 | | | VUS |
| 5 | 10:115356904 | c.4371del, p.(Thr1458Glnfs*36) | Frameshift | 37/42 | 100 | | | P |
| | 10:115355414 | c.4504C>T, p.(Arg1502*) | Nonsense | 38/42 | 95 | | | P |
| **6** | **10:115423570** | **c.72G>C, p.(Gln24His)** | **Missense** | **1/42** | **34** | **Delet. (0.01)** | **Full** | **P** |
| | **10:115355414** | **c.4504C>T, p.(Arg1502*)** | **Nonsense** | **38/42** | **95** | | | **P** |
| 7 | 10:115423570 | c.72G>C, p.(Gln24His) | Missense | 1/42 | 34 | Delet. (0.01) | Full | P |
| | 10:115355414 | c.4504C>T, p.(Arg1502*) | Nonsense | 38/42 | 95 | | | P |
| **8** | **10:115356904** | **c.4371del, p.(Thr1458Glnfs*36)** | **Frameshift** | **37/42** | **100** | | | **P** |
| **9** | **10:115374675** | **c.3109C>T, p.(Arg1037*)** | **Nonsense** | **28/42** | **2** | | | **LP** |
| 10 | 10:115364570 | c.4025G>A, p.(Ser1342Asn) | Missense | 35/42 | 3 | Delet. (0.01) | Full | VUS |
| | 10:115423640 | c.2T>C, p.(Met1?) | Start lost | 1/42 | 21 | | | LP |
| **11** | **10:115400070** | **c.1344T>A, p.(Tyr448*)** | **Nonsense** | **14/42** | **35** | | | **P** |

Genomic coordinates refer to human reference genome (GRCh37/hg19) and mutation nomenclature is based on GenBank accession NM_001261463.1 (NRAP). Homozygotes and compound heterozygous patients are marked with bold font. Cons, Conservation in mammals; Delet., Deleterious; GnomAD AC, Allele count in gnomAD reference population cohort; LP, Likely pathogenic; P, Pathogenic; VUS, Variant of Uncertain Significance. No homozygotes for these variants are present in the gnomAD reference population cohort.

## Enrichment of *NRAP* variants in DCM

We identified cases with two rare *NRAP* variants, of which at least one was a PTV in 11 out 577 (1.91%) patients in the DCM group but none were in either the non-DCM cardiac group or control group (Table 2). Frequency of such variant combination was significantly greater in the DCM group vs. controls (OR 1052, 95%CI 62–17876, p<0.0001; Table 3). Three of the

**Table 3. Enrichment of rare *NRAP* variants in patients with dilated cardiomyopathy (DCM).**

| | Control group | Non-DCM cardiac group | OR (95% CI), P-value | DCM group | OR (95% CI), p-value |
|---|---|---|---|---|---|
| Individuals (n) | 25912 | 5150 | | 577 | |
| | | | **Dominant hypothesis** | | |
| Only one PTV variant | 75 (0.29%) | 24 (0.47%) | 1.61 (1.02–2.56), p = 0.04 | 11 (1.91%) | 6.71 (3.5–12.7), p<0.0001 |
| Only one missense variant | 698 (2.45%) | 132 (2.56%) | 1.05 (0.86–1.26), p = 0.65 | 10 (1.74%) | 0.70 (0.4–1.3), p = 0.27 |
| | | | **Recessive hypothesis** | | |
| Two missense variants | 27 (0.10%) | 1 (0.02%) | 0.18 (0.02–1.35), p = 0.10 | 0 (0.0%) | 0.81 (0.05–13.4), p = 0.89 |
| One PTV + one missense | 0 | 0 | NA | 6 (1.04%) | 590 (33–10494), p<0.0001 |
| Two PTV variants | 0 | 0 | NA | 5 (0.87%) | 407 (22–7575), p<0.0001 |
| PTV + missense or 2 PTVs | 0 | 0 | NA | 11 (1.91%) | 1052(62–17876), p<0.0001 |

Control group patients underwent genetic testing due to non-cardiac reasons and non-DCM group patients due to cardiological reasons excluding patients with DCM or suspected DCM. DCM group includes patients tested with a DCM Panel or other panels because of confirmed or suspected DCM. Abbreviations: pts, patients; OR, odds ratio; 95% CI, 95% confidence interval, DCM, dilated cardiomyopathy, PTV, Protein-truncating variant (means here nonsense, frameshift variant, consensus splice site, start lost). Only variants with 100 or fewer heterozygous individuals in a gnomAD reference population cohort were included in calculations.

**Table 4. Clinical characteristics of patients with biallelic or potentially biallelic *NRAP* variants.**

| Patient | Age | Gender | Phenotype | LVEDD(mm)/EF% | Age at onset | Htx | Died | Other |
|---------|-----|--------|-----------|---------------|--------------|-----|------|-------|
| 1 | 19 | F | DCM | NA | <19 | Yes, at age of 19 | | |
| 2 | 36 | F | DCM | 71/10-15% | NA | | 36 | |
| 3 | 31 | M | DCM | 70/13% | 28 | | | Mild LGE |
| 4 | 57 | M | DCM | NA | 56 | | | Maximum treatment for heart failure |
| 5 | 4 | F | DCM | 58/20% | 4 | LVAD | | |
| 6 | 59 | F | DCM | 63/34% | 46 | | | |
| 7 | 59 | M | DCM | NA/30% | NA | | | LBBB |
| 8 | 43 | M | DCM | NA | 22 | Yes, at age 43 | | |
| 9 | 53 | F | DCM | NA | NA | | | History of cardiac arrest |
| 10 | 48 | M | DCM/HF | NA | NA | | | |
| 11 | 2 | M | DCM | NA | NA | | 2 | |

Age means age at last follow-up. Abbreviations: DCM, Dilated cardiomyopathy; EF, Ejection fraction; F, Female; HF, Heart failure; Htx, Heart transplantation; LBBB, Left bundle branch block; LGE, Late gadolinium enhancement at cardiac MRI; LVEDD, Left ventricular end-diastolic diameter in mm; M, Male; NA, Not available or not known.

patients had familial cardiomyopathy and eight had a sporadic disease. In these 11 individuals, four had a homozygous PTV, one had two heterozygous PTVs (phase unknown) and two were compound heterozygous for a PTV and a missense variant (*in trans*). In five patients, the phase of the *NRAP* variants was unknown. Two presumably unrelated patients had the same frameshift/missense variant combination (c.4371del, p.Thr1458Glnfs*36 and c.72G>C, p.Gln24His) and two had the same nonsense/missense variant combination (c.4504C>T, p.Arg1502* and c.72G>C, p.Gln24His). Thus, the p.(Gln24His) missense variant was observed altogether in four presumably unrelated patients. This variant may in fact lead to splicing defect as it affects the last nucleotide of the exon 1. Alamut Visual Splicing software v2.11 (Interactive Biosoftware, France) predicts that this variant either leads to loss of the native splice donor (NNSPLICE) or significant weakening of this site (SSF, MaxEnt). One patient had a start lost variant, *NRAP* p.(Met1?), which is expected to cause loss-of-function as there is an alternative out-of-frame start codon 5-bp down-stream from the wild type initiation codon.

None of these 11 patients had an alternative molecular diagnosis identified in either large NGS panel (n = 9) or exome sequencing (n = 2). Six (55%) of the patients have had major end-points, defined as history of cardiac transplantation (n = 2), death on waiting list for heart transplantation (n = 1) or during left ventricular assist device (LVAD) treatment (n = 2), and previous cardiac arrest (n = 1). The mean age at the time of the major endpoint was 22.8±19.4 years (Table 4). Four of these six patients had homozygous PTV in *NRAP* and one patient had two PTVs (phase unknown). Patients with two PTVs (n = 5) were younger at disease onset than patients with PTV + missense variant (n = 6) combination (19.6±20.4 vs. 48.3±12.3 years, p = 0.018). None of the patients had known skeletal muscle involvement.

A single heterozygous PTV without another rare *NRAP* variant was observed in 11 patients (1.91%) and they were also enriched in the DCM group (OR 6.71, 95% CI 3.5–12.7, p<0.0001; Table 3). The single heterozygous PTV group excludes all patients with two rare *NRAP* variants as defined earlier. However, one of these patients also had another moderately rare (500 heterozygotes in gnomAD) missense variant in *NRAP* (c.2963A>C, p.(Gln988Pro); phase unknown) in addition to a start-lost variant. The patient had no alternative molecular diagnosis in established cardiomyopathy genes. Of the 11 patients with only one heterozygous PTV in *NRAP*, three had another molecular diagnosis including three PTVs affecting A-band of *TTN* and one had an additional frameshift variant in *DSP*.

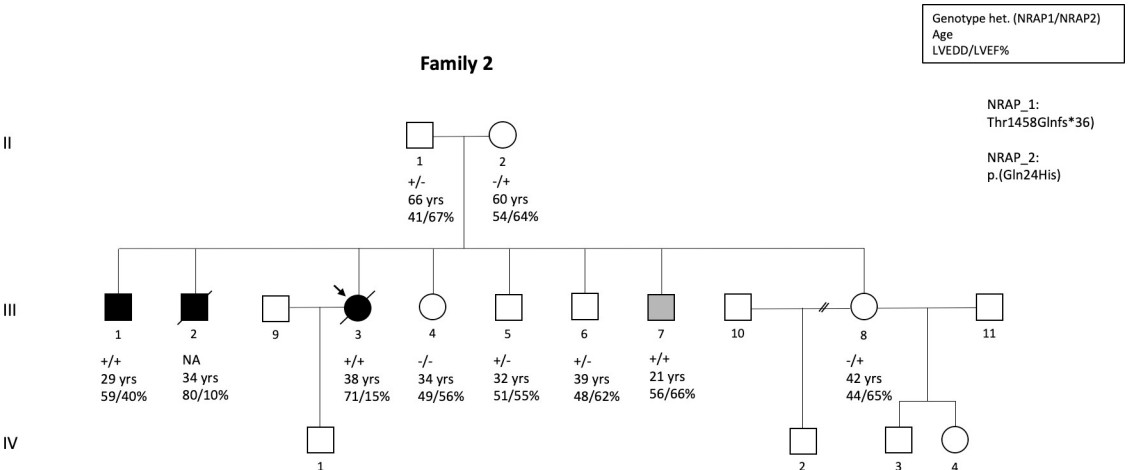

**Fig 1.** Pedigree of the family-2 where the index patient and her affected brother were compound heterozygous for c.4371del, p.(Thr1458Glnfs*36) and c.72G>C, p.(Gln24His) in *NRAP* similarly as her 21-year brother who were on medication initiated before the results of genetic testing were available due to borderline imaging findings suggesting cardiomyopathy. He did not fulfill diagnostic criteria of DCM at the time of the study. DNA was not available from one affected individual who died for DCM at age of 34. All family members who were heterozygous only for the other variant or were homozygous for the wild type allele were unaffected.

### Familial segregation

We were able to recruit two out of the three probands with familial disease and one with sporadic disease for additional screening. Co-segregation was assessed from 18 family members who underwent screening of familial variants and clinical history, as well as and clinical evaluation including ECG and echocardiography. Cardiac MRI was performed as needed.

In family 2, the proband died at the age of 38 years from severe biventricular heart failure. She was compound heterozygous for c.4371del, p.(Thr1458Glnfs*36) and c.72G>C, p.(Gln24His) in *NRAP* (Table 2, Fig 1). At the time of last imaging study, her LVEDD was 71 mm, LVEF was 13% and RVEF was 17%, and she had elevated levels of TnI and proBNP and a widened QRS (132 ms). One of the proband's brothers was diagnosed with DCM at the age of 24 years and he died at age of 34 years of severe biventricular heart failure. No DNA sample was available from this individual for genetic testing. Two of the family members were compound heterozygous for the same variants. One had a diagnosis of mild DCM at age 20 and no progression since initiating ACE inhibitor treatment, and the other had upper normal LV size despite of treatment initiation at the age of 21 years. All five heterozygous siblings and one with wild type allele were healthy. The parents of the proband were both heterozygous for one the variants and had normal echocardiography.

In family 6, the proband was diagnosed with DCM at the age of 47 years due to dilated LV and reduced LV function (LVEDD 63mm, EF 34%). She was compound heterozygous for c.4504C>T, p.(Arg1502*) and c.72G>C, p.(Gln24His) in *NRAP* (Table 2, Fig 2). Mild improvement in LV size and function were observed with medical treatment. The proband's sister died at the age 40 years due to DCM. She was an obligate compound heterozygote for the same variants as the proband, which was discovered after the testing of her children. Three of the proband's siblings have died during childhood, but no samples were available from any of them for genetic testing. In the extended family, two heterozygous individuals and two with wild type alleles were healthy. The proband's parents, who were both obligatory heterozygotes for one variant, had no known cardiomyopathy and had a normal life span.

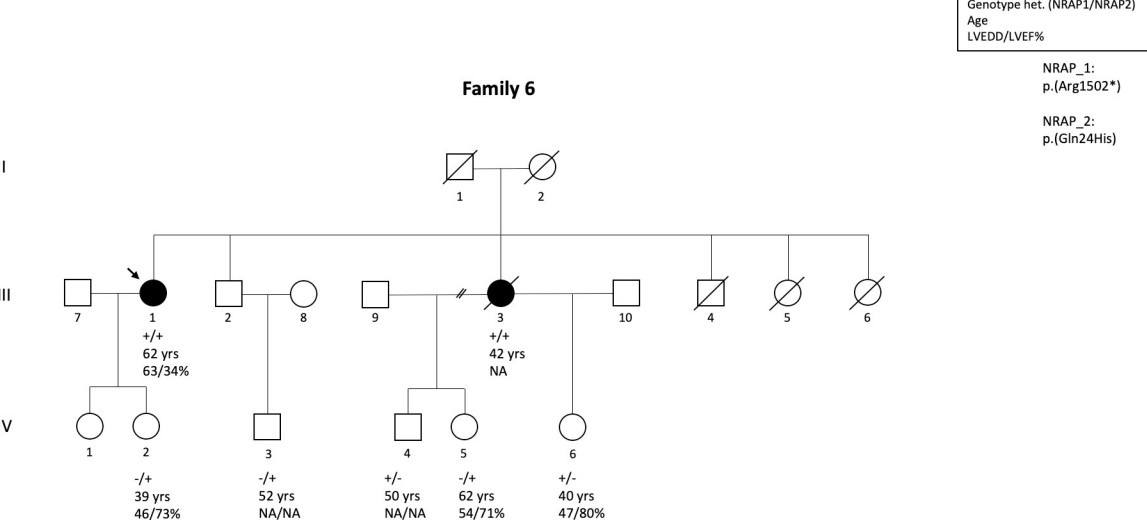

**Fig 2. Pedigree of the family-6 where the index patient and her affected brother were compound heterozygous for c.4504C>T, p.(Arg1502\*) and c.72G>C, p.(Gln24His) in *NRAP*.** All family members who were heterozygous only for the other variant were unaffected.

In family 11, the proband was diagnosed with DCM at age 2 due to dilated LV and reduced LV function. The patient was homozygous for c.1344T>A, p.(Tyr448\*) in *NRAP* (Table 2, Fig 3). He received LVAD soon after hospitalization due to severe heart failure but he died before planned transplantation. The proband's parents were heterozygotes for the variant and his older sister was homozygous for a wild type allele. All family members were healthy.

## Estimating frequency of biallelic protein truncating *NRAP* variants in general population

As we discovered significant new evidence supporting the role of biallelic *NRAP* variants in DCM, we decided to further estimate the potential contribution of this gene on DCM at a global scale. We queried the count of *NRAP*-PTV in gnomAD reference population v2.1.1. In total 733 high-quality PTVs were present in the database. The average number of alleles reported at these positions was 233,756 indicating a cumulative allele frequency of 0.31%.

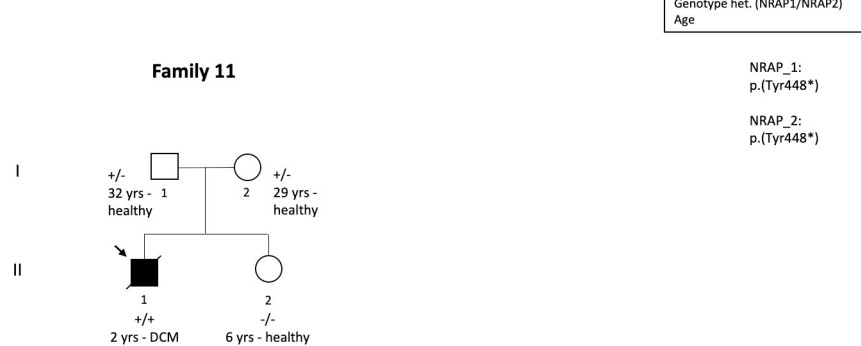

**Fig 3. Pedigree of the family-11 where the index patient was homozygous for c.1344T>A, p.(Tyr448\*) in *NRAP* whereas all other family members who were heterozygous for the variant or had homozygous wild type allele were unaffected.**

Thus, the probability of homozygosity or compound heterozygosity is approximately 0.000983% at the individual level. This is equal to 1 case per 101,700 individuals if we assume that only PTV variants would be disease causing.

## Discussion

Our data suggest that the variants in the *NRAP* gene are associated with DCM and may explain up to 1.91% of DCM cases in an unselected clinical cohort consisting of patients with either clinically diagnosed DCM or suspected DCM. Because small disease cohorts may provide inaccurate estimates due to non-random inclusion and pure coincidence, we decided to estimate the prevalence of potentially biallelic *NRAP*-PTV in a large population data set (gnomAD). This analysis yielded a frequency of 0.000983%, equal to 1 case per 101,700 individuals. If all of these variants were fully penetrant, *NRAP* might explain up to 0.34%-2.03% of all DCM cases when relying on variable (1:3,000 to 1:500) estimates of DCM prevalence in the general population. Thus, our DCM patient cohort and population data cohort provide essentially similar estimates of the contribution of *NRAP* in DCM.

In general, non-syndromic familial cardiomyopathies follow dominant inheritance [1,18]. In 2016, the *ALPK3* gene was discovered to be a rare cause of a recessive pediatric cardiomyopathy, which typically presents with DCM and non-compaction that progress to hypertrophic cardiomyopathy and possibly some syndromic features [19]. Some other genes such as *GATAD1* [20], *PLEKHM2* [21], and *PPCS* [22] have been shown to associate with recessive non-syndromic DCM. However, not much evidence has been gathered after initial reports, possibly reflecting the rarity of such gene-to-phenotype associations. Classic genes encoding cardiac desmosome proteins initially connected to ARVC/arrhythmogenic cardiomyopathy are now considered established causes of the DCM phenotype [10,23]. Notably, some variants in *DSG2* gene cause recessive ARVC that may be difficult to distinguish from DCM [24]. However, based on the numbers of reported patients and mutation database submissions (e.g. ClinVar) of patients carrying variants in previously described recessive cardiomyopathy genes, it seems likely that *NRAP* has a more prominent contribution to the etiology.

Previous reports involving *NRAP* gene did not include segregation analysis [13] had insufficient data obtained from the family studies to fully support segregation or were inconsistent with co-segregation [12,14,15]. In the first study suggesting association of *NRAP* with cardiomyopathy, the proband's 35-year-old brother who was homozygous for PTV in *NRAP* was considered unaffected while being asymptomatic and having normal echocardiography and ECG [12]. Thus, the authors concluded that *NRAP* may be a low penetrance genetic risk factor for DCM even though the previous observation can also be explained by age-dependent penetrance of cardiomyopathies. Later, Ahmed et al. published a consanguineous pedigree in which the index patient was a baby girl who presented at the age of 13 months with heart failure, easy fatigability, weakness, irritability, and shortness of breath and was diagnosed with DCM [15]. Whole exome sequencing revealed that her healthy 33-year-old father was homozygous for the same frameshift variant identified in the proband whereas the mother was heterozygous. The proband's family history included one stillbirth and another brother who was diagnosed with cardiomyopathy at the age of 12 months and died at 17 months without a molecular diagnosis (samples were not available for genetic testing). Otherwise the extended pedigree did not reveal any known cardiomyopathy cases, which also suggests recessive inheritance. Our previous study reported a family in which the index patient who was diagnosed with DCM at the age of 3 years [14]. The proband was homozygous for *NRAP* p.(Tyr448*). Three family members were heterozygous for the variant and one had a homozygous wildtype allele, and all of them were considered healthy. Individuals who are heterozygous for a single

LoF variant in *NRAP* are cardiologically healthy in all previously published reports as well as in our study suggesting that *NRAP* does not cause dominantly inherited monogenic DCM. However, we cannot exclude the possibility that it would increase susceptibility to cardiomyopathy even when heterozygous due to observed enrichment of single LoF variants in our DCM cohort.

*NRAP* seems to associate with severe DCM as the proportion of patients with major cardiac endpoint (death, cardiac arrest, transplant and LVAD) is similar or higher compared to *LMNA* related cardiac laminopathy (54.5% vs. 58.3%) [25]. *NRAP* patients also seem to have an earlier onset of major cardiac end-points when compared to cardiac laminopathy or DCM in general (22.8±19.4 vs. 51.0±8.7 and was 59.0±14.2 years) [25]. In addition, the rate of cardiac transplantation and LVAD utilization was higher in our *NRAP* group compared to a Norwegian *LMNA* cohort (45% vs. 33%) [26].

Our data also suggest that two PTVs in *NRAP* cause more severe disease than PTV + missense combination in *NRAP*. There are no previous observations on the PTV + missense variant combination in DCM, thus further studies are needed to confirm whether the previous assumption is correct. Given that our data did not show enrichment of potentially biallelic missense variants in the DCM group, these variants may not contribute to the DCM pathogenesis alone. However, at this time we cannot exclude the possibility that a small proportion of biallelic missense variants are disease causing alone.

*NRAP* appear to play important role in myocardial architecture and sarcomere function, supporting the biological plausibility of our findings. The *NRAP* gene on chromosome 10q25.3 encodes the nebulin related anchoring protein. This protein is involved in anchoring terminal actin filaments to the membrane, tension transmission from myofibrils to extracellular matrix, as well as having a significant role in myofibrillogenesis during cardiomyocyte development, and it is involved in the sarcomeric contraction cycle in adult heart [27,28]. The N-terminal LIM domain of NRAP interacts with α-actinin and talin [29,30], while the domain with single repeats interacts also with actin, the Kelch-like family member 41 (*KLHL41*) [31], and cysteine and glycine-rich protein 3 (*CSRP3*) [27], and the C-terminal super repeats interact with filamin C (*FLNC*) [31] and vinculin (*VCL*) [29]. Experimentally, upregulation of NRAP expression was observed in DCM mice models and human DCM patients [27,32]. This has been suggested to be an adaptive response to correct for disorganized actin thin filament architecture at intercalated disc junctions. NRAP is expressed in the myocardium and in striated muscle. Truszkowska et al. previously reported an absence of NRAP protein in the myocardium of a DCM proband with biallelic PTV in *NRAP* whileNRAP protein was clearly present in a control heart [12].

## Study limitations

In one of the three probands with familial DCM, we were unable to obtain samples from the parents and other family members to further prove segregation of the phenotype with the genotype. Similarly, DNA samples were available only in one of the eight probands with sporadic DCM. Even though the data supported recessive inheritance since heterozygous individuals were unaffected, more thorough segregation studies would have brought depth to the scientific message especially by clarifying penetrance of *NRAP* related DCM. Moreover, no functional studies were carried out, nor animal models were generated for any of the identified variants. Four of the patients carried the same splice region missense variant, *NRAP* p. (Gln24His), but we did not perform transcriptional analysis to determine this variant's effect on splicing that would have increase our understanding on disease mechanisms. None of the *NRAP* variants detected via NGS were confirmed with Sanger sequencing since all of them had

high variant call quality score, fulfilled several other quality control criteria for true positive call, and the reporting followed CLIA/CAP/ISO-15189 approved policy. This study provides the first statistical association between the *NRAP* gene and DCM without mechanistic insights or evidence that have been partially provided in the initial case reports.

The results of this study demonstrate significant enrichment of *NRAP* variants in DCM patients with severe clinical events and their co-segregation in multiple families support an inclusion of *NRAP* in genetic testing of cardiomyopathies.

## Supporting information

**S1 File.**
(XLSX)

## Acknowledgments

We thank all the healthcare professional sending their samples to Blueprint Genetics for genetic testing and providing clinical information on patients' phenotype that enabled this registry study. We also thank index patients and their family members who were recruited to either the Inherited Cardiomyopathies Study or KidCMP Study for the participation.

## Author Contributions

**Conceptualization:** Juha W. Koskenvuo, Johanna Tommiska, Sini Weckström, Tiina H. Ojala, Tero-Pekka Alastalo, Tiina Heliö.

**Data curation:** Juha W. Koskenvuo, Inka Saarinen, Saija Ahonen, Johanna Tommiska, Eija H. Seppälä, Sari Tuupanen, Tiia Kangas-Kontio, Jennifer Schleit, Krista Heliö, Julie Hathaway, Anders Gummesson, Pia Dahlberg, Tiina H. Ojala, Ville Vepsäläinen, Ville Kytölä, Mikko Muona, Johanna Sistonen, Pertteli Salmenperä, Massimiliano Gentile, Jussi Paananen, Samuel Myllykangas, Tero-Pekka Alastalo, Tiina Heliö.

**Formal analysis:** Juha W. Koskenvuo, Inka Saarinen, Saija Ahonen, Johanna Tommiska, Sini Weckström, Eija H. Seppälä, Sari Tuupanen, Tiia Kangas-Kontio, Jennifer Schleit, Krista Heliö, Julie Hathaway, Anders Gummesson, Pia Dahlberg, Tiina H. Ojala, Ville Vepsäläinen, Ville Kytölä, Mikko Muona, Johanna Sistonen, Pertteli Salmenperä, Massimiliano Gentile, Jussi Paananen, Samuel Myllykangas, Tero-Pekka Alastalo, Tiina Heliö.

**Funding acquisition:** Juha W. Koskenvuo, Sini Weckström, Tiina Heliö.

**Investigation:** Juha W. Koskenvuo, Inka Saarinen, Saija Ahonen, Johanna Tommiska, Eija H. Seppälä, Sari Tuupanen, Tiia Kangas-Kontio, Jennifer Schleit, Krista Heliö, Julie Hathaway, Anders Gummesson, Pia Dahlberg, Tiina H. Ojala, Ville Vepsäläinen, Ville Kytölä, Mikko Muona, Johanna Sistonen, Pertteli Salmenperä, Massimiliano Gentile, Jussi Paananen, Samuel Myllykangas, Tero-Pekka Alastalo, Tiina Heliö.

**Methodology:** Juha W. Koskenvuo, Inka Saarinen, Saija Ahonen, Johanna Tommiska, Sari Tuupanen, Tiia Kangas-Kontio, Jennifer Schleit, Krista Heliö, Julie Hathaway, Ville Kytölä, Mikko Muona, Johanna Sistonen, Pertteli Salmenperä, Massimiliano Gentile, Jussi Paananen, Samuel Myllykangas, Tero-Pekka Alastalo, Tiina Heliö.

**Validation:** Inka Saarinen.

**Writing – original draft:** Juha W. Koskenvuo, Tiina Heliö.

**Writing – review & editing:** Inka Saarinen, Saija Ahonen, Johanna Tommiska, Sini Weck-
ström, Eija H. Seppälä, Sari Tuupanen, Tiia Kangas-Kontio, Jennifer Schleit, Krista Heliö,
Julie Hathaway, Anders Gummesson, Pia Dahlberg, Tiina H. Ojala, Ville Vepsäläinen, Ville
Kytölä, Mikko Muona, Johanna Sistonen, Pertteli Salmenperä, Massimiliano Gentile, Jussi
Paananen, Samuel Myllykangas, Tero-Pekka Alastalo.

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
