## [Decision Letter · Decision Letter 0]

31 Jul 2020

PONE-D-20-18740

Biallelic loss-of-function in NRAP is a common cause of recessive dilated cardiomyopathy

PLOS ONE

Dear Dr. Koskenvuo,

Thank you for submitting your manuscript to PLOS ONE. After careful consideration, we feel that it has merit but does not fully meet PLOS ONE’s publication criteria as it currently stands. Therefore, we invite you to submit a revised version of the manuscript that addresses the points raised during the review process.

1. Address reviewer 1's concern on the lack of detail on the clinical phenotyping of patients and include more details on these.

2. Provider more details for the pipeline used in the bioinformatics analysis.

3  Address reviewer's comments on improvements for Tables 1 and 2.

4. Fix discrepancies in numbers identified by reviewer 2.

5. Respond to reviewer 4's comments related to exome sequencing and lack of detection of other DCM genes.

6. Consider adding in the reference suggested by Reviewer 4.

7.  Provide greater detail on the inclusion and exclusion criteria for variants.  See comments from reviewers.  Also address how these variants meet ACMG criteria.

8.  Address the other points raised by the reviewers.

We look forward to receiving your revised manuscript.

Kind regards,

Amanda Ewart Toland, Ph.D.

Academic Editor

PLOS ONE

Journal Requirements:

"Drs. Koskenvuo, Saarinen, Ahonen, Tommiska, Seppälä, Tuupanen, Kangas-Kontio, Schleit, Hathaway, Kytölä, Muona, Sistonen, Salmenperä, Gentile, Paananen, Myllykangas, Alastalo are full-time employees of Blueprint Genetics, which offers genetic diagnostics for cardiomyopathies. All other authors have reported that they have no relationships relevant to the contents of this paper to disclose."

We note that one or more of the authors are employed by a commercial company: Blueprint Genetics, a Quest Diagnostics Company.

2.1. Please provide an amended Funding Statement declaring this commercial affiliation, as well as a statement regarding the Role of Funders in your study. If the funding organization did not play a role in the study design, data collection and analysis, decision to publish, or preparation of the manuscript and only provided financial support in the form of authors' salaries and/or research materials, please review your statements relating to the author contributions, and ensure you have specifically and accurately indicated the role(s) that these authors had in your study. You can update author roles in the Author Contributions section of the online submission form.

2.2. Please also provide an updated Competing Interests Statement declaring this commercial affiliation along with any other relevant declarations relating to employment, consultancy, patents, products in development, or marketed products, etc. 

Reviewers' comments:

Reviewer's Responses to Questions

**Comments to the Author**

1. Is the manuscript technically sound, and do the data support the conclusions?

Reviewer #1: Yes

Reviewer #2: Yes

Reviewer #3: Partly

Reviewer #4: Partly

2. Has the statistical analysis been performed appropriately and rigorously? 

Reviewer #1: Yes

Reviewer #2: Yes

Reviewer #3: I Don't Know

Reviewer #4: Yes

3. Have the authors made all data underlying the findings in their manuscript fully available?

Reviewer #1: No

Reviewer #2: Yes

Reviewer #3: Yes

Reviewer #4: Yes

4. Is the manuscript presented in an intelligible fashion and written in standard English?

Reviewer #1: Yes

Reviewer #2: Yes

Reviewer #3: No

Reviewer #4: Yes

5. Review Comments to the Author

Reviewer #1: In this manuscript by Koskenvuo and colleagues utilize a large genetic testing database to identify a number of rare variants in NRAP associated with DCM. Overall, this is an appropriately concise report that is well written. It attempts to tackle a common, yet challenging problem – what is the cause of genotype negative DCM?

The size of the cohort analyzed, and the limited co-segregation present are strengths of this paper. However, there are a number of points which should be expanded or clarified in my opinion.

In the introduction, a brief explanation of the physiologic role of NRAP would be helpful. Is it expressed in the heart? Solely in the heart? Would its expression pattern lend itself to a cardiac myocardium only disease?

A major limitation of this manuscript is the lack of clinical phenotyping of the patients with DCM. This should be addressed. What is the evidence that the cohort that was screened had non-ischemic, congenital DCM? This is even more of any issue for the NRAP-positive individuals.

The bioinformatic pipeline used to identify these variants is not well delineated. How were other variants that localized to genes which are known to cause DCM excluded as possibilities?

Were copy number variants identified? This is important to address as there are CNVs associated with DCM (1p36 deletion syndrome) which may confound this data.

The variant inclusion and exclusion data is unclear. For example, having an exclusionary criteria which appears to be <100 individuals is not appropriate as coverage in gnomAD is variable. Thus, a poorly covered area with fewer individuals sequenced may thus have fewer minor allele positive individuals that are nonetheless a high minor allele frequency. Minor allele frequency should be used.

Inclusion criteria for missense variants are particularly unclear. What criteria is used to determine whether a missense variant is included?

Table 1 seems to be extraneous and seems odd. I think it should be removed and potentially referenced in the Discussion

What was the pathogenicity assessment of the NRAP variants by 2015 ACMG guidelines?

Reviewer #2: Koskenvuo et al. presented interesting novel data on NRAP gene’s implication in the pathogenesis of autosomal recessive dilated cardiomyopathy (DCM).

Authors reviewed genetic results of 577 patients either with a confirmed or suspected diagnosis of DCM and compared them with data of control cohort of 31062 individuals, including both non-cardiac controls (n>25000) and non DCM cardiac controls (n>5000).

They identified two rare variants in NRAP gene [(four homozygous truncating (PTV), or compound heterozygous (one patient had two different PTV and 6 had PTV/missense)] in 11 unrelated probands with DCM (1.9%) but none in the controls, with OR of 1052 in comparison to controls. They concluded that NRAP variants could explain 0.25%-2.46% of all DCM cases.

Also, Authors stress the fact that NRAP patients seem to have an earlier onset of major cardiac end-points compared to cardiac laminopathy or DCM, and higher requirement for cardiac transplantation/LVAD.

This is a very important, well designed and rigorously performed study, rather nicely presented.

However, page 10:

Authors state:

“In these 11 individuals four had a homozygous PTV, one had two heterozygous PTVs and two were compound heterozygous for a PTV/missense variant.” This adds up to 7 , and not to 11.

Although later we can learn that the same variants were present in other patients, and the Table 2 lists them, it is unclear and should be corrected.

Page 19

Authors state:

"In the first study suggesting association of NRAP with cardiomyopathy, the proband’s 35-year-old brother who was compound heterozygous for two PTVs in NRAP”. This is untrue. As Authors correctly state in the Table 1, the first identified variant was homozygous nonsense NRAP variant (p.Arg1502*).

Reviewer #3: Koskenvuo et al report a study where they identified 4 cases with biallelic NRAP protein truncating variants (PTV), 2 cases with NRAP compound heterozygous PTV and a missense variant, 1 case with two PTVs (phase unknown) and 4 cases with a PTV and a missense variant (phase unknown), in a large dilated cardiomyopathy cohort (n=577). The aim was to assess the association of bi-allelic loss of function (LoF) variants in NRAP with autosomal recessive cardiomyopathy, since previously, only 4 such cases have been reported. Large disease cohorts are a useful resource to assess for the presence of any gene’s variants and to study their association with the disease. This study provides supports the role of NRAP biallelic loss of function variants and their association with autosomal recessive cardiomyopathy, though with reduced penetrance. The data presented in the study does not support for a causal role of NRAP LoF variants in autosomal dominant dilated cardiomyopathy. However, this manuscript in its current form needs substantial revision before it is suitable for publication in PLos One. This report can benefit from addition of phenotype details of the affected individuals and further discussion on incomplete penetrance or age related penetrance etc.

Major comments:

In this study the authors found 6 cases with biallelic NRAP variants (4 homozygous & 2 with PTV and a missense variant in trans/compound heterozygous). This needs to be clearly stated in the text as the other 5 sets of NRAP variants may or may not be in trans.

Were the variants Sanger confirmed in the 11 cases ? If so, please mention that in the methods.

Table 1

• It is best to list the variants in the “chr-genomic coordinate- Ref allele- Alt allele” format. This format is most useful along with the HGVS nomenclature for the cDNA , protein and the coding exon to which the LoF variants maps to (for example 3/20).

• Please add the units and reference ranges for the CK & ProBNP levels listed

• Addition of the gender of the affected individual will also be helpful

• In the footnotes, it needs to be mentioned that homozygotes for these variants in Gnomad were not seen.

• Please mention the NRAP NM_ID used by the published reports listed in this table.

Table 2

• Table 2 contains information on the cases reported in the current study; however, the organization of this table is not reader friendly.

• Please reword the legend for clarity and make it concise

Suggested Legend “NRAP variants identified in individual with confirmed or suspected dilated cardiomyopathy”

• It is best to have one row each for one affected individual

• Please add available phenotype information for each individual including gender, age of onset, CK levels, available cardiac biopsy

• It is best to list the variants in the “chr-genomic coordinate- Ref allele- Alt allele” format

• List the variants in Hgvs nomenclature (cDNA and protein)

• Add a column to clarify whether it is compound heterozygous or phase unknown

• The cases where unambiguous biallelic NRAP variants were seen (6/11) should be listed first

• the number of coding exon where the variant is located will be a useful addition especially, for the C terminal LoF variants, to assess whether they are predicted to undergo NMD or will result in truncation

• The “type of variant” column is redundant and may be removed

• SIFT and conservation data can be listed in the footnotes

• In the footnotes, it needs to be mentioned that homozygotes for these variants in Gnomad were not seen

For calculation of the population based prevalence using the frequency of NRAP LoF variants in Gnomad v.2.1, were the NRAP LoF variants seen in the last coding exon also included? The LOF variants in the last coding exon of NRAP may not undergo NMD and may result in a truncated product. Hence if these were included, the prevalence estimate derived for NRAP associated recessive cardiomyopathy might be a slight overestimate. Further, there were two high quality homozygous LoFs seen for NRAP in gnomad v.2.1.1 – were they included/removed for the prevalence calculation?

Discussion

If the age of onset data is available for the 6 individuals, along with the 4 published reports the heterogeneity seen should be discussed. If a large variability is seen, it might be supportive of age related penetrance. In addition, the age of onset or its variability, if seen for other AR non syndromic dilated cardiomyopathy genes, is also worth a mention.

Since NRAP is predominantly expressed in the heart & skeletal muscle, in the 6 cases with bialleleic variants were there any muscle issues seen in the affected individuals? Were any muscle issues noted in the 4 previously published cases? If so, this needs to be mentioned in the discussion

The authors list a handful of genes, which cause AR non-syndromic cardiomyopathy. Is age related penetrance or incomplete penetrance a feature seen for these genes as well ? in individuals who carry biallelic variants in these genes. Or this is uniquely seen for NRAP?

This will be insightful; Further, Gnomad V.2.1.1 data has two high quality homozygous LoF variants in NRAP. The authors should attempt to provide a potential explanation for this observation.

It is interesting that a single missense variant (p.Gln24His) which could potentially affect splicing is seen in 4 affected individuals.

Minor comments

Several sentences lack clarity and I urge the authors to read through the manuscript carefully and edit the manuscript for clarity and typos.

For example,

Title

“ Biallelic loss-of-function in NRAP is a common cause of recessive dilated cardiomyopathy”

Instead the following title is clearer

“ Biallelic loss-of-function variants in NRAP are a common cause of recessive dilated cardiomyopathy”

Abstract

“Confirmed or potentially biallelic NRAP variants were enriched in DCM”

the usage of the word confirmed is confusing ; instead it is suggested to use “Biallelic or potential bi-allelic”

“Biallelic (n=6) or two (n=5) NRAP variants (two PTVs or PTV+missense) were”

(two PTVs or PTV+missense)

Instead the following addition will make it more clear.

“Biallelic (n=6) or two (n=5) NRAP variants (two PTVs or PTV+missense) were”

(two PTVs or PTV+missense, phase unknown )

Methods: Patients

Please clearly mention whether the HQSA also included NRAP.

Page 5

estimated int the range of 1:500 to 1:3,000 [2–4].

Page 10:

“Visual Splicing software v2.11 (Interactive Biosoftware, France) predicts that this variant either leads to either loss of the native splice donor”

Instead

“Visual Splicing software v2.11 (Interactive Biosoftware, France) predicts that this variant either leads to loss of the native splice donor”

“Enrichment of NRAP variants in DCM

We identified two rare NRAP variants, of which at least one was a PTV in 11 out”

Instead

We identified cases, which carried two rare NRAP variants, of which atleast one was a PTV

Gene names should be italicized in the manuscript text.

Reviewer #4: The authors identified biallelic loss-of-function in NARP as a common cause of recessive dilated cardiomyopathy with next generation sequencing. However, there are some issues to address.

Major issues:

1. The study didn’t detect the variants of well-established causal genes of DCM. Is the DCM phenotype caused by biallelic loss-of-function in NARP or variants of the known causal genes, such as TTN, RBM20, et, al? It would better to perform whole exome sequencing or targeted next generation sequencing with causal genes of DCM on the cohort.

2. It would be better to confirm the variants in NARP by Sanger sequencing, especially in family segregation analysis due to the false positive rate of next generation sequencing.

Minor issues:

1. Vasilescu et, al reported NRAP underlying childhood dilated cardiomyopathy. It would be better to mention their work in discussion section. (J Am Coll Cardiol. 2018 Nov 6;72(19):2324-2338.) in NARP.

6. PLOS authors have the option to publish the peer review history of their article (what does this mean?). If published, this will include your full peer review and any attached files.

Reviewer #1: **Yes: **Andrew Landstrom

Reviewer #2: No

Reviewer #3: No

Reviewer #4: No

---

## [Author Response · Author response to Decision Letter 0]

17 Nov 2020

All comments raised by the editor and reviewers have now addressed.

---

## [Decision Letter · Decision Letter 1]

16 Dec 2020

PONE-D-20-18740R1

Biallelic loss-of-function in NRAP is a common cause of recessive dilated cardiomyopathy

PLOS ONE

Dear Dr. Koskenvuo,

Thank you for submitting your manuscript to PLOS ONE. After careful consideration, we feel that it has merit but does not fully meet PLOS ONE’s publication criteria as it currently stands. Therefore, we invite you to submit a revised version of the manuscript that addresses the points raised during the review process.

1.  Soften language suggesting that biallelic NRAP loss is a common cause of AR DCM throughout manuscript and in title.

2.  Expand description of bioinformatics analysis of NGS data in the methods.

3. Include details of inclusion/exclusion criteria for study population.  See reviewer's comments.

4. Consider doing Sanger to confirm the 11 variants found.  If Sanger is not done, list lack of confirmation of these variants as a study limitation in the Discussion.

5.  Check for spelling errors.  See reviewer's comments.

We look forward to receiving your revised manuscript.

Kind regards,

Amanda Ewart Toland, Ph.D.

Academic Editor

PLOS ONE

Reviewers' comments:

Reviewer's Responses to Questions

**Comments to the Author**

1. If the authors have adequately addressed your comments raised in a previous round of review and you feel that this manuscript is now acceptable for publication, you may indicate that here to bypass the “Comments to the Author” section, enter your conflict of interest statement in the “Confidential to Editor” section, and submit your "Accept" recommendation.

Reviewer #1: (No Response)

Reviewer #2: All comments have been addressed

Reviewer #4: (No Response)

2. Is the manuscript technically sound, and do the data support the conclusions?

Reviewer #1: Yes

Reviewer #2: Yes

Reviewer #4: Yes

3. Has the statistical analysis been performed appropriately and rigorously? 

Reviewer #1: I Don't Know

Reviewer #2: Yes

Reviewer #4: Yes

4. Have the authors made all data underlying the findings in their manuscript fully available?

Reviewer #1: (No Response)

Reviewer #2: Yes

Reviewer #4: Yes

5. Is the manuscript presented in an intelligible fashion and written in standard English?

Reviewer #1: Yes

Reviewer #2: Yes

Reviewer #4: Yes

6. Review Comments to the Author

Reviewer #1: The revisions to this manuscript appear reasonable and the manuscript is greatly improved. I have one lingering concern.

Overall, I think the language associating biallelic loss of NRAP is too strong. For example, the title stating that it is the most common cause of AR DCM, the conclusion in the abstract and discussed in the paper is outside the scope of the data. Given the heterogeneity of the genetic substrate of DCM, additional, independent validation studies are needed to corroborate this finding. This should be highlighted in the manuscript, including in the title, and the language appropriately softened.

Reviewer #2: The manuscript is sound, concise and interesting. Although clinical data are not robust, the report nicely underlies severe clinical endpoints related to the biallelic loss-of-function NRAP variants causing autosomal recessive DCM.

Spelling mistakes:

fatigability

wildtype

Reviewer #4: The authors found that rare NRAP variants were associated with the pathogenesis of autosome recessive dilated cardiomyopathy through genetic testing performed on a large cohort. Moreover, co-segregation analysis provided a strong supporting evidence for this conclusion.

However, there are some issues to address.

1. Have the authors performed Sanger sequencing to validate the 11 variants as NGS possess relatively high error rate?

2. Please described the bioinformatics analysis in Materials and methods section in details.

3. It is better to list out the inclusion and exclusion criteria of the recruited subjects. How to define a DCM patient and what kind of individuals can be classified as suspected DCM instead of non-DCM cardiac group?

7. PLOS authors have the option to publish the peer review history of their article (what does this mean?). If published, this will include your full peer review and any attached files.

Reviewer #1: **Yes: **Andrew Landstrom

Reviewer #2: No

Reviewer #4: No

---

## [Author Response · Author response to Decision Letter 1]

4 Jan 2021

PONE-D-20-18740R1

Therefore, we invite you to submit a revised version of the manuscript that addresses the points raised during the review process.

(Ed1.1). Soften language suggesting that biallelic NRAP loss is a common cause of AR DCM throughout manuscript and in title.

RESPONSE: <<New title: Biallelic loss-of-function in NRAP is a cause of recessive dilated cardiomyopathy>>.

(Ed1.2). Expand description of bioinformatics analysis of NGS data in the methods.

RESPONSE: <<We have added information on bioinformatics pipeline into methods section as much as possible. Most of the detailed settings are proprietary, and are thus intentionally left out.>>

(Ed1.3). Include details of inclusion/exclusion criteria for study population. See reviewer's comments.

RESPONSE: << This information is now available in two places at methods section:

1) The patients paragraph: The inclusion criteria for DCM group (see later) was referral to genetic testing due to diagnosis or clinical suspicion of DCM. 

2) NRAP variants paragraph: Frequency of such high-quality variants were compared between patients with clinical or suspected dilated cardiomyopathy (DCM group), other cardiology indication (Non-DCM cardiac group consisting patients tested due diagnosis or suspicion inherited aortopathy, channelopathy or cardiomyopathy other than DCM) or any other clinical indication for panel or exome testing (Control group).>>.

(Ed1.4). Consider doing Sanger to confirm the 11 variants found. If Sanger is not done, list lack of confirmation of these variants as a study limitation in the Discussion.

RESPONSE: <<We decided to continue without Sanger confirmation and added the required comment into the limitations section. To our opinion, this request would have been appropriate 7-10 years ago and in other circumstranmces. We have performed all tests in accredited laboratory by qualified personnel, fully documented workflows, validated assays, CLIA lab sign-off process etc. This is very different world compared to research laboratory.>>

(Ed1.5). Check for spelling errors. See reviewer's comments.

RESPONSE: <<Done>>.

Reviewer #1: The revisions to this manuscript appear reasonable and the manuscript is greatly improved. I have one lingering concern.

(Rev1.1). Overall, I think the language associating biallelic loss of NRAP is too strong. For example, the title stating that it is the most common cause of AR DCM, the conclusion in the abstract and discussed in the paper is outside the scope of the data. Given the heterogeneity of the genetic substrate of DCM, additional, independent validation studies are needed to corroborate this finding. This should be highlighted in the manuscript, including in the title, and the language appropriately softened.

RESPONSE: <<We have modified the title and conclusions throughout the manuscript as requested>>.

Reviewer #2: 

(Rev2.1).The manuscript is sound, concise and interesting. Although clinical data are not robust, the report nicely underlies severe clinical endpoints related to the biallelic loss-of-function NRAP variants causing autosomal recessive DCM.

Spelling mistakes:

fatigability

wildtype

RESPONSE: <<Done>>.

Reviewer #4: 

(Rev4.1).The authors found that rare NRAP variants were associated with the pathogenesis of autosome recessive dilated cardiomyopathy through genetic testing performed on a large cohort. Moreover, co-segregation analysis provided a strong supporting evidence for this conclusion.

However, there are some issues to address.

1. Have the authors performed Sanger sequencing to validate the 11 variants as NGS possess relatively high error rate?

RESPONSE: <<We decided to continue without Sanger confirmation and added the required comment into the limitations section. >>

(Rev4.2). Please described the bioinformatics analysis in Materials and methods section in details.

RESPONSE: <<We have added information on bioinformatics pipeline into methods section as much as possible. Most of the detailed settings are proprietary, and are thus intentionally left out.>>

(Rev4.3). It is better to list out the inclusion and exclusion criteria of the recruited subjects. How to define a DCM patient and what kind of individuals can be classified as suspected DCM instead of non-DCM cardiac group? 

RESPONSE: << This information is now available in two places at methods section:

1) The patients paragraph: The inclusion criteria for DCM group (see later) was referral to genetic testing due to diagnosis or clinical suspicion of DCM. 

2) NRAP variants paragraph: Frequency of such high-quality variants were compared between patients with clinical or suspected dilated cardiomyopathy (DCM group), other cardiology indication (Non-DCM cardiac group consisting patients tested due diagnosis or suspicion inherited aortopathy, channelopathy or cardiomyopathy other than DCM) or any other clinical indication for panel or exome testing (Control group).>>.

---

## [Editor Report · Decision Letter 2]

6 Jan 2021

Biallelic loss-of-function in NRAP is a cause of recessive dilated cardiomyopathy

PONE-D-20-18740R2

Dear Dr. Koskenvuo,

We’re pleased to inform you that your manuscript has been judged scientifically suitable for publication and will be formally accepted for publication once it meets all outstanding technical requirements.

Kind regards,

Amanda Ewart Toland, Ph.D.

Academic Editor

PLOS ONE
---

## [Editor Report · Acceptance letter]

12 Jan 2021

PONE-D-20-18740R2 

Biallelic loss-of-function in *NRAP* is a cause of recessive dilated cardiomyopathy 

Dear Dr. Koskenvuo:

I'm pleased to inform you that your manuscript has been deemed suitable for publication in PLOS ONE. Congratulations! Your manuscript is now with our production department. 

Kind regards, 

on behalf of

Dr. Amanda Ewart Toland 

Academic Editor

PLOS ONE